# Granulosa secreted factors improve the developmental competence of cumulus oocyte complexes from small antral follicles in sheep

Shiva Rouhollahi Varnosfaderani, Mehdi Hajian, Farnoosh Jafarpour, Faezeh Ghazvini Zadegan, Mohammad Hossein Nasr-Esfahani [ID] *

Department of Reproductive Biotechnology, Reproductive Biomedicine Research Center, Royan Institute for Biotechnology, ACECR, Isfahan, Iran

* mh.nasr-esfahani@royaninstitute.org

**Data Availability Statement:** All relevant data are within the manuscript text, tables and figures.

## Abstract

Oocyte *in vitro* maturation can be improved by mimicking the intra-follicular environment. Oocyte, cumulus cells, granulosa cells, and circulating factors act as meiotic regulators in follicles and maintain oocyte in the meiotic phase until oocyte becomes competent and ready to be ovulated. In a randomized experimental design, an ovine model was used to optimize the standard *in vitro* maturation media by Granulosa secreted factors. **At first, the** development capacity of oocyte derived from medium (>4 to 6 mm) and small (2 to ≤4 mm) size follicles was determined. Differential gene expression of granulosa secreted factors and their receptors were compared between the cumulus cells of the two groups. Then, the best time and concentration for arresting oocytes at the germinal vesicle stage by natriuretic peptide type C (CNP) were determined by nuclear staining in both groups. Oocyte quality was further confirmed by calcein uptake and gene expression. The developmental competence of cumulus oocyte complexes derived from small size follicles that were cultured in the presence of CNP in combination with amphiregulin (AREG) and prostaglandin E2 (PGE2) for 24 h was determined. Finally, embryo quality was specified by assessing expressions of *NANOG*, *SOX2*, *CDX2*, *OCT4*, and *TET1*. The cumulus oocyte complexes derived from small size follicles had a lower capacity to form blastocyst in comparison with cumulus oocyte complexes derived from medium size follicles. Prostaglandin E receptor 2 and prostaglandin-endoperoxide synthase 2 had significantly lower expression in cumulus cells derived from small size follicles in comparison with cumulus cells derived from medium size follicles. Natriuretic peptide type C increased the percentage of cumulus oocyte complexes arresting at the germinal vesicle stage in both oocytes derived from medium and small follicles. Gap junction communication was also improved in the presence of natriuretic peptide type C. In oocytes derived from small size follicles; best blastocyst rates were achieved by sequential exposure of cumulus oocyte complexes in [TCM+CNP (6 h), then cultured in TCM+AREG+PGE2 (18h)] and [TCM+CNP (6 h), then cultured in conventional IVM supplements+AREG+PGE2 (18h)]. Increased *SOX2* expression was observed in [TCM+CNP (6 h), then cultured in TCM+AREG+PGE2 (18h)], while decreased OCT4 expression was

**Funding:** The authors received no specific funding for this work.

**Competing interests:** The authors have declared that no competing interests exist.

observed in [TCM+CNP (6 h), then cultured in conventional IVM supplements+AREG +PGE2 (18h)]. It seems that the natriuretic peptide type C modulates meiotic progression, and oocyte development is probably mediated by amphiregulin and prostaglandin E2. These results may provide an alternative IVM method to optimize *in vitro* embryo production in sheep and subsequently for humans.

## Introduction

Perhaps the greatest challenge of assisted reproductive techniques is *in vitro* maturation. Edwards reported the first IVM in humans, and Cha showed the first live birth after IVM in a woman with premature ovarian failure (POF). Since then, a great result has been achieved by capacitation pre-maturation-IVM (CAPA-IVM). Despite this recent progress in CAPA-IVM, this method has not replaced the conventional IVM procedure in humans and other species [1–3].

Improved development of *in vitro* matured oocytes can only be achieved by expanding our knowledge regarding the complex dialog between the oocyte and its surrounding somatic cells within follicle [4,5]. Nevertheless, this has proven to be challenging because: 1) dealing with a heterogeneous population of oocytes harvested from follicles of different size (2–6 mm in sheep and goat, 2–8 mm in bovine and < 12mm in human) is a formidable issue; 2) cumulus oocyte complexes (COCs) derived from small antral follicles possess less competence in response to regulatory and specific ligands because of an immature signaling capacity [6–10]; and 3) Isolation of COCs from their natural follicular environment results in spontaneous mei-otic progression and, thus, asynchronization of nuclear and cytoplasmic maturation [11].

Recent microarray analyses between developmentally competent and incompetent COCs identified differential expression of quality marker genes in human and bovine [12–15]. Based on these studies, critical deficiencies in IVM may be related to a lack of granulosa cell-COC communication. The well-known factors secreted from granulosa cells (GCs) during *in vivo* maturation process are natriuretic peptides (NPs), epidermal growth factor (EGF)-like factors, and prostaglandins (PGs) that regulated extracellular cellular matrix, metabolism and immune system [12,16–19].

Members of the NP peptides include atrial natriuretic peptide (ANP), brain natriuretic peptide (BNP), and c-type natriuretic peptide (CNP). They are secreted by granulosa cells [20]. CNP is considered as the main NP and binds to natriuretic peptide receptor 2 (NPR2) on CCs, inducing the production of cyclic guanosine monophosphate (cGMP). Cyclic GMP enters oocyte via GJC and regulates levels of cyclic adenosine monophosphate (cAMP) by suppress-ing the hydrolyzing activity of oocyte-specific phosphodiesterases 3A (PDE3A). Increased cAMP level maintains meiotic arrest of immature oocytes within follicles [14, 21–23]. Besides, it has been stated that estradiol can mediate the expression of *NPR2* on CCs [22].

EGF-like factors, AREG, epiregulin (EREG), and betacellulin (BTC) act on EGF receptor (EGFR) and activate the extracellular signal-regulated protein kinases 1 and 2 (ERK1/2), protein kinase C (PKC) pathways and other signaling pathways in granulosa and cumulus cells (CCs), which make the COCs competent to respond to Luteinizing hormone (LH) surge [17, 24]. One of the target genes of the ERK1/2 pathway is prostaglandin synthase 2 (*PTGS2*), which leads to the production of PGE2 through a positive feedback loop between AREG and PGE2. Prostaglandin E2 (PGE2), an arachidonic acid-derived lipid mediator, is an autocrine/paracrine factor that medi-ates gonadotrophin (Gn) stimulation of cumulus expansion and oocyte maturation [16, 25–28].

Culture of COCs in medium containing granulosa secreted factors has been suggested to increase the proximity of *in vitro* culture to *in vivo* condition and improve the efficiency of *in*

*vitro* oocyte maturation, especially in COCs derived from small follicles (2 to ≤ 4mm) with lower developmental competence compared to follicles of ≥ 4 to 6mm or of greater size [29, 30]. Our goal is to select ovine as a model for human IVM in future studies.

To achieve this aim, we initially exposed COCs from small follicles to CNP for 6 h and then to TCM 199 as a base of routine culture medium in the presence of AREG and/or PGE2, and the results of each group were compared with their corresponding group.

## Materials and methods

All procedures were approved by the Institutional Review Board of the Royan Institute (the ethical guidelines established by the Institutional Ethical Committee of the Royan Institute). The majority of chemicals and media were obtained from Sigma Chemical Co. (St. Louis, MO, USA) and Gibco (Grand Island, NY, USA), respectively. Other chemicals used for specific experiments from other companies are cited in the text as required.

### *In vitro* maturation (IVM)

Ovine ovaries from a local abattoir were transported to the laboratory at 6:00 pm in saline (15˚C–20˚C) and stored for an additional 12 h at 15˚C since the lab working hours start in the morning [31, 32]. COCs were isolated from medium (≥ 4 to 6 mm) and small (2 to ≤ 4 mm) size follicles with the aid of 21-G needles. COCs with more than three layers of cumulus cells and homogenous cytoplasm were washed with the HEPES tissue culture medium 199 (HTCM199) + 1 mg/mL PVA+ 4 mg/mL BSA. Finally, 50 COCs were cultured in 400 µl containing 1 mg/mL PVA+ 8 mg/mL BSA in each group, according to experimental design (Fig 1) in 4 well dishes without mineral oil for 24 h at 38.5˚C and 5% $CO_2$ in the air [33, 34].

### Nuclear status

COCs were treated with 300 IU/mL hyaluronidase and vortexes to remove CCs. Denuded oocytes (DOs) were subsequently fixed for 20 minutes in 4% paraformaldehyde. To visualize chromatin, DOs were stained with Hoechst 33342 (10 µg/mL) for 5 minutes. After mounting, images of stained oocytes were captured and assessed by high-resolution digital camera (DP-72 Olympus, Japan) using DP2-BSW software [35]. The percentage of oocytes at the GV stage in each group was determined.

### Cumulus expansion index (CEI)

After 24 h IVM, CEI was scored on a 0 to 4 scale, as described by Vanderhyden et al., 1999: score 0, no expansion; score 1, no CC expansion but cells appear as spherical; score 2, only the outermost layers of CCs have expanded; score 3, all cell layers have expanded except the corona radiata; and score 4, expansion has occurred in all cell layers including the corona radiate [36].

### *In vitro* fertilization (IVF)

Domestic sheep breed, Rouge de l'ouest, was used for *in vitro* fertilization. Motile sperms were separated using a swim-up preparation; 100 µl of fresh sperm (from a ram of proven fertility) was kept under Tyrode's albumin lactate pyruvate medium in 5% $CO_2$, 38.5˚C, and humidified air for up to 45 minutes to allow motile sperm to swim up. Subsequently, insemination was carried out by adding $5 \times 10^3$ sperm/ matured COCs in fertilization medium (NaCl 114 mM, KCl 3.15 mM, $NaH_2PO_4$ 0.39 mM, Na-lactate 13.3 mM, $CaCl_2$ 2 mM, $MgCl_2$ 0.5 mM, Na-pyruvate 0.2 mM, Penicillin 50 IU/ml, Streptomycin 50 µg/ml, $NaHCO_3$ 25 mM, Heparin

## Experimental design

| | Goal | Base medium | Groups | Assessments | |
|---|---|---|---|---|---|
| **Part I** | Comparison of follicles with two different size | Conventional IVM | Oocyte derived from small and medium size follicles | *In vitro* development | Fig. 2 |
| | | | | Genes expression | Fig. 3 |
| **Part II** | Meiotic Regulatory | TCM | **CNP (0, 10, 100,1000 nM)** <br> 1) TCM, <br> 2) E2, <br> 3) CNP, <br> 4) CNP+E2 | Meiotic arrest | Fig. 4 |
| | | | | | Fig. 5 |
| | | | 1) TCM, <br> 2) CNP | Genes expression and gap junctions communications | Fig. 6 |
| **Part III** | Developmental Regulatory | TCM | **AREG (50, 100,300 nM)** | CEI, MII, Development | Table 2 |
| | | | **PGE2 (0.1, 1, 10 µM)** | | Table 3 |
| | | | 1) TCM, <br> 2) CNP→TCM, <br> 3) CNP→AREG[*], <br> 4) CNP→PGE2, <br> 5) CNP→AREG+PGE2[*], <br> 6) CNP→Conventional IVM+AREG+PGE2[*] | *In vitro* development | Fig. 7 |
| | | | 1) TCM, <br> 2) CNP→AREG+PGE2 | | Fig. 8 |
| | | | 1) TCM, <br> 2) Conventional IVM[†], <br> 3) CNP→Conventional IVM+AREG+PGE2, <br> 4) CNP→AREG+PGE2 | Genes expression | Fig. 9 |

**Symbol and abbreviations description**

→: transferred to.
IVM: *in vitro* maturation; TCM: tissue culture medium; CNP: c-type natriuretic peptide; E2: estradiol; AREG: Amphiregulin; PGE2: Prostaglandin E2; CEI: Cumulus expansion index; MII: Metaphase II.
[*] The group's names in the text were defined based on the group numbers in part III.
[†] Conventional IVM contains TCM + LH (10µg/ml) + FSH (10µg/ml) + IGF (100ng/ml) + EGF (10ng/ml) + Cysteamine (0.1mM) + FBS (15%) + E2 (1µg/ml). COCs are cultured in conventional IVM for 0-24h.

**Fig 1. Experimental design.**

10 µg/ml, 6 mg/ml BSA) for 20 h at 38.5˚C under 5% $CO_2$ in humidified air overlaid with light mineral oil. On the next day, to remove the cumulus cells, the presumptive zygotes were vortexed in HTCM199 + 1mg/ml PVA+ 4mg/ml BSA for 3 minutes [37]. Then, they were cultured for 8 days in BO-IVC (Brackett-Oliphant *in vitro* culture) medium (Bioscience, UK) at 39˚C, 6% $CO_2$, 5% $O_2$ in humidified air under mineral oil. Day 0 was defined as the day of insemination. Therefore, cleavage and blastocyst rates (over cleavage) were determined on days 3 and 7 post embryo culture.

## Relative gene expression

In each group, at the desired time, CCs were collected from COCs after vortexing for 4 minutes. Then, CCs were separated from oocytes, and CCs were stored in RLT buffer at -70˚C until RNA extraction. RNA was extracted from CCs with the aid of an RNeasyMini Kit (Qiagen) followed by DNase I (Fermenas; EN0521) treatment. Total RNA (1000 ng) was reverse-transcribed using a Takara cDNA Synthesis kit (Takara; #RR07A) according to the manufacturer's instructions. The primers were designed using Beacon and Oligo7 softwares. Efficiency correction for each primer was performed by serial dilution of positive control cDNA as a template. Please note that we are comparing the level of expression within different periods; we have no control group. Therefore, the transcripts abundance of 6 genes (*AREG*, *EGFR*, *NPPC*, *NPR2*, *PTGER2*, *PTGS2*, *GJA4*, and *GJA1*) (Table 1) was normalized to B-actin as reference gene using $2^{-(\text{delta CT})}$ rather than using $2^{-(\text{delta delta CT})}$.

Total RNA of five blastocysts in each group that was stored in RLT buffer at -70˚C was extracted with the aid of Micro- RNeasy kit (Qiagen, Canada). For reverse transcription, 14 µl of total RNA (1µg) was used in a final volume of 20 µl reaction that contained 1 µl of random hexamer, 2 µl RT buffer (10x), 1 µl of RNase inhibitor (40 IU), 1 µl of reverse transcriptase (Takara; #RR07A), and 1 µl DEPC water. Reverse transcription was carried out at 37˚C for 15 minutes, followed by 85˚C for 5 seconds. Moreover, real-time PCR was implemented using 1 µl of cDNA (50 ng), 5 µl of the SYBR Green qPCR Master Mix (2x) (Fermentas, Germany) and 1 µl of forward and reverse primers (5 pM) adjusted to a total volume of 10 µl using nuclease-free water. Real-time PCR program was 1) 95˚C 4 min, 2) 94˚C 10 s, 3) Ta 30 s, and 4) 72˚C 30 s, for 40 cycles. The transcripts abundance of 5 genes (nanog homeobox (*NANOG*), SRY-Box transcription factor 2 (*SOX2*), caudal type homeobox 2 (*CDX2*), octamer-binding transcription

**Table 1. Primer sequences.**

| Gene symbol | Forward primer (5´-3´) | Reverse primer (5´-3´) | Annealing temp. (˚C) |
|---|---|---|---|
| *OCT4* | GGAAAGGTGTTCAGCCA | ATTCTCGTTGTTGTCAGC | 57 |
| *NANOG* | CCTCTCAACATACAGCC | TCTTATTGGACTCATTACC | 51 |
| *SOX2* | GAGAACAATGACACACCAA | TGCTGAAATGAGGAGGAG | 57 |
| *CDX2* | CCCCAAGTGAAAACCAG | TGAGAGCCCCAGTGTG | 56 |
| *TET1* | CGGAAGAAAGAAGGTCGTC | GAATAACACCAAATGAGCGG | 57 |
| *AREG* | ATACTGCTGGATTAGATG | CTGTGGTTCATTATCATAC | 49 |
| *EGFR* | ACAAGACAATAAGCCACTT | CACCCAAAGGAGAGAAAG | 50 |
| *NPPC* | CCAATCTCAAGGACGAC | TTGGACAAACCCTTCTT | 53 |
| *NPR2* | AACTCCACTCTCAACTCTG | CTCTGAATTGCCGAACTG | 56 |
| *PTGS2* | CTTCCAGCCGCAGTAG | GGCATCTATGTCTCCGTA | 58 |
| *PTGER2* | CATCCTGAGACCTCCTGTTC | CTACCACTTCTTAACTACCATCCT | 58 |
| *GJA4* | TCCTTCCTAATGACCAGAG | GTAAGTTGTCTCCGAATCC | 53 |
| *GJA1* | GTGTCGTTGGTGTCTCTTG | CAGTGGTAGTGTGGTAAGGA | 61 |
| *β-actin* | CCATCGGCAATGAGCGGT | CGTGTTGGCGTAGAGGTC | 58 |

factor 4 (*OCT4*) and ten-eleven translocation methylcytosine dioxygenase 1 (*TET1*)) (Table 1) were normalized to beta-actin as reference gene, and $2^{\wedge-(\text{delta delta CT})}$ was presented.

## Statistical analysis

Data percentages were modeled to a normal distribution by ArcSin transformation. Cleavage, blastocysts rates, and relative gene expression results among more than three groups were examined using a one-way ANOVA followed by Tukey's post hoc tests (Figs 3–5, Fig 7 and Fig 9). The t-test was used for cleavage, blastocysts rates, and relative gene expression data between two groups (Fig 2, Fig 6 and Fig 8). The differences were considered significant at P<0.05. All results were presented as means ± standard error of the mean (SEM).

## Results

### Comparison of developmental competence and gene expression between COCs derived from medium (>4 to 6 mm) and small (2 to ≤4 mm) size follicles

The COCs derived from small size follicles had a lower capacity to form blastocyst in comparison with COCs derived from medium size follicles [16.6 ± 5.0 vs. 42.9 ± 3.5; p<0.05; Fig 2; see the experimental design (Fig 1, part I)].

No significant difference was found for *AREG* gene expression between CCs derived from two groups at 3 time points. *EGFR* gene expression in CCs derived from medium size follicles at 12 h post-maturation was significantly higher than small size follicles (p<0.05). No significant difference was found for *NPPC* gene expression between CCs derived from two groups at 3 time points. *NPR2 gene* expression significantly declined during the first 12 h of maturation in CCs derived from small size follicles (p<0.05). Expressions of *PTGER2* and *PTGS2 genes* between two groups at 3 time points were lower in CCs derived from small size follicles, except for *PTGER2* that had higher expression in small compared with medium size follicles at 0h post-maturation (p<0.05, Fig 3).

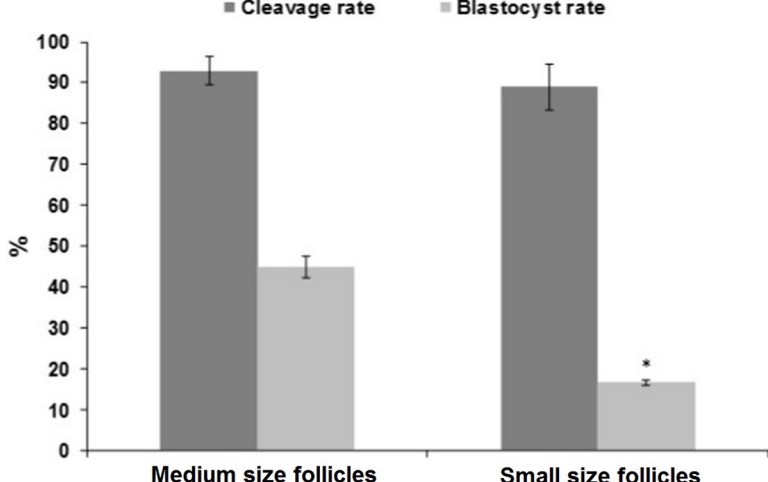

**Fig 2. Developmental competency of *in vitro* conventional maturation of COCs derived from medium (>4 to 6 mm) compared to small (2 to ≤4 mm) size follicles.** 5 replicates and minimum number of oocytes in each replicate were 50. Culture medium was conventional maturation medium containing TCM + LH (10μg/ml) + FSH (10μg/ml) + IGF (100ng/ml) + EGF (10ng/ml) + Cys (0.1mM) + FBS (15%) + E2 (1μg/ml). The asterisk represents a significant difference (P<0.05) within columns with the same color.

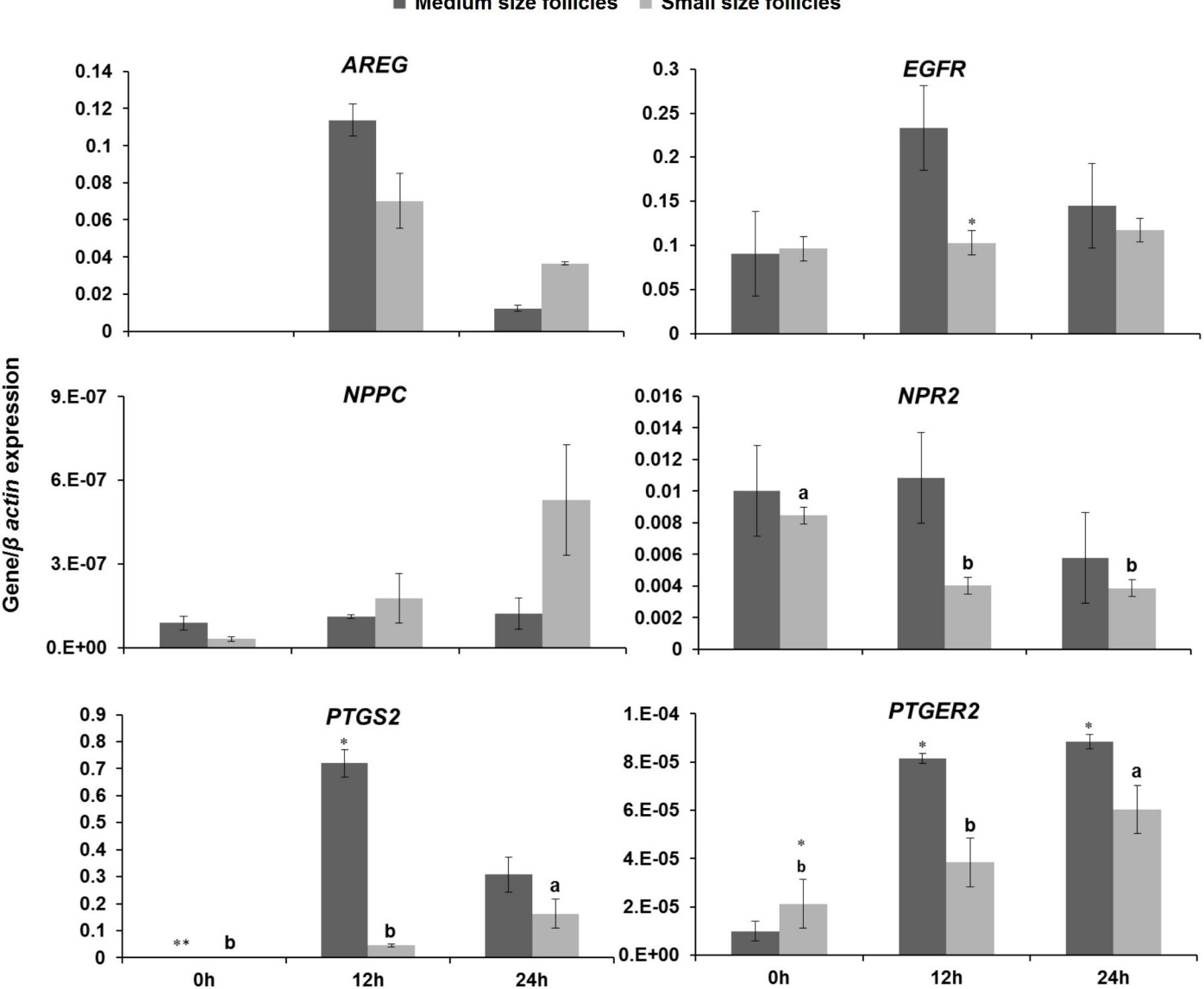

**Fig 3. Gene expression of** *AREG, EGFR, NPPC, NPR2, PGE2,* **and** *PTGS2* **between cumulus cells derived from medium (>4 to 6 mm) and small (2 to ≤4 mm) size follicles at 0, 12 and 24 h post-** *in vitro* **maturation in a conventional medium.** 3 replicates and minimum number of cumulus cells in each replicate were $2\times10^{5}$. Different superscripts demonstrate significant differences (P<0.05) within columns with the same color. The asterisks represent significant differences between the two groups at the same time of maturation. $2^{\wedge-(\text{delta CT})}$ was presented for gene expression. *AREG*: amphiregulin, *NPPC*: natriuretic peptide precursor C, *PTGS2*: prostaglandin-endoperoxide synthase 2, *EGFR*: epidermal growth factor receptor, *NPR2*: natriuretic peptide receptor B, *PGE2*: prostaglandin E synthase 2.

## Effects of CNP on meiotic arrest, relative gene expression, and gap junction's communications

As the concentration of CNP increased (10, 100, 1000 nM), the percentage of oocytes derived from medium follicles remaining at the GV stage increased (P<0.05). In another group, there was no significant difference between 100 and 1000 nM of CNP (P>0.05, Fig 4). According to these results, concentrations of 1000 nM and 100 nM for CNP were selected for both medium and small size follicles, respectively [Fig 4, see the experimental design (Fig 1, part II)].

Meiotic progression of oocytes derived from medium follicles occurred around 6 h after the onset of IVM in TCM. The percentage of oocytes derived from medium size follicles at the GV

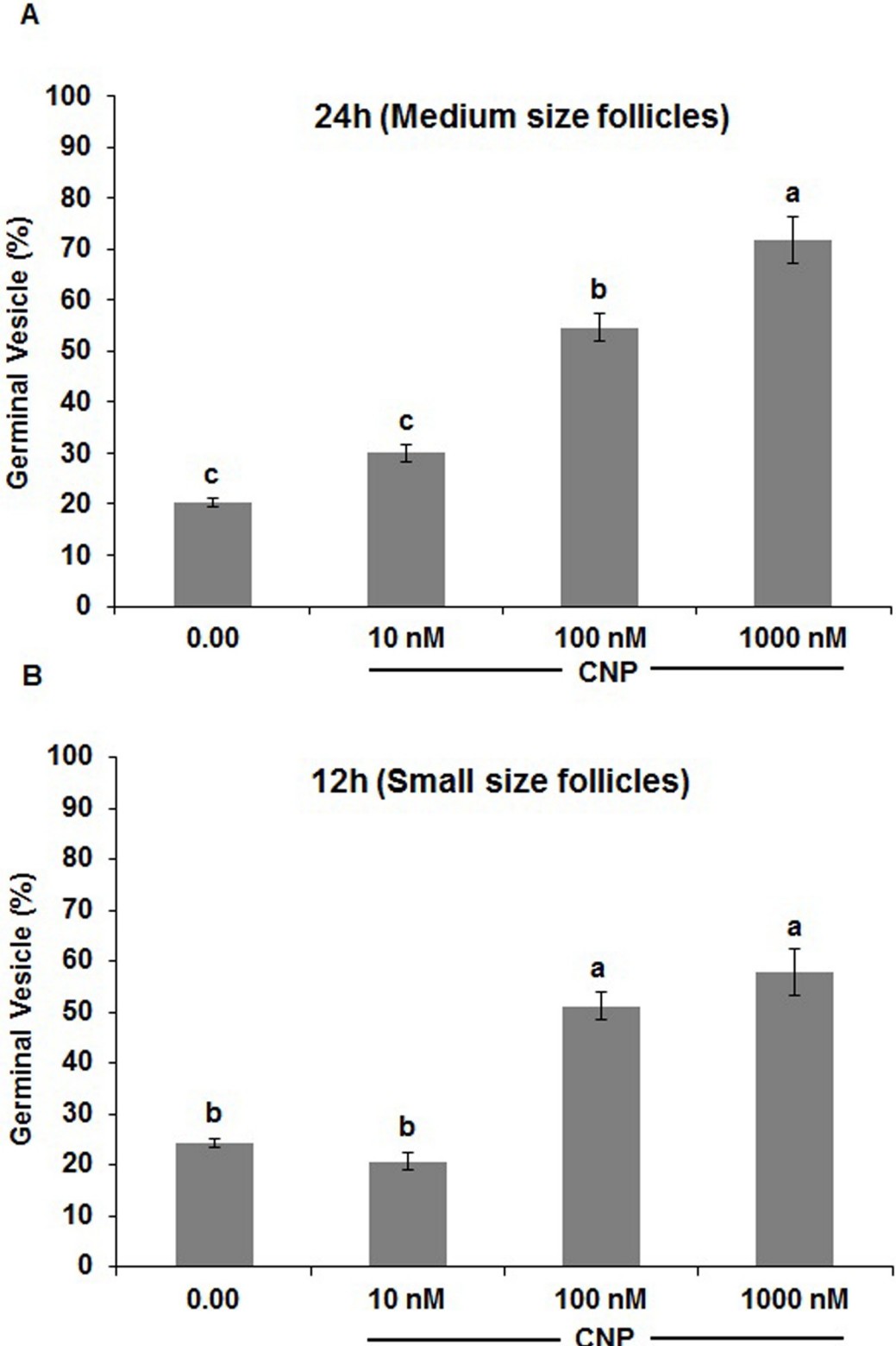

**Fig 4. Determination of optimal concentration of CNP between oocytes derived from medium (>4 to 6 mm) and small (2 to ≤4 mm) size follicles.** 3 replicates and the minimum number of oocytes in each replicate were 30. Different superscripts demonstrate significant differences (P<0.05). CNP: natriuretic peptide type C.

**A**
**Medium size follicles**

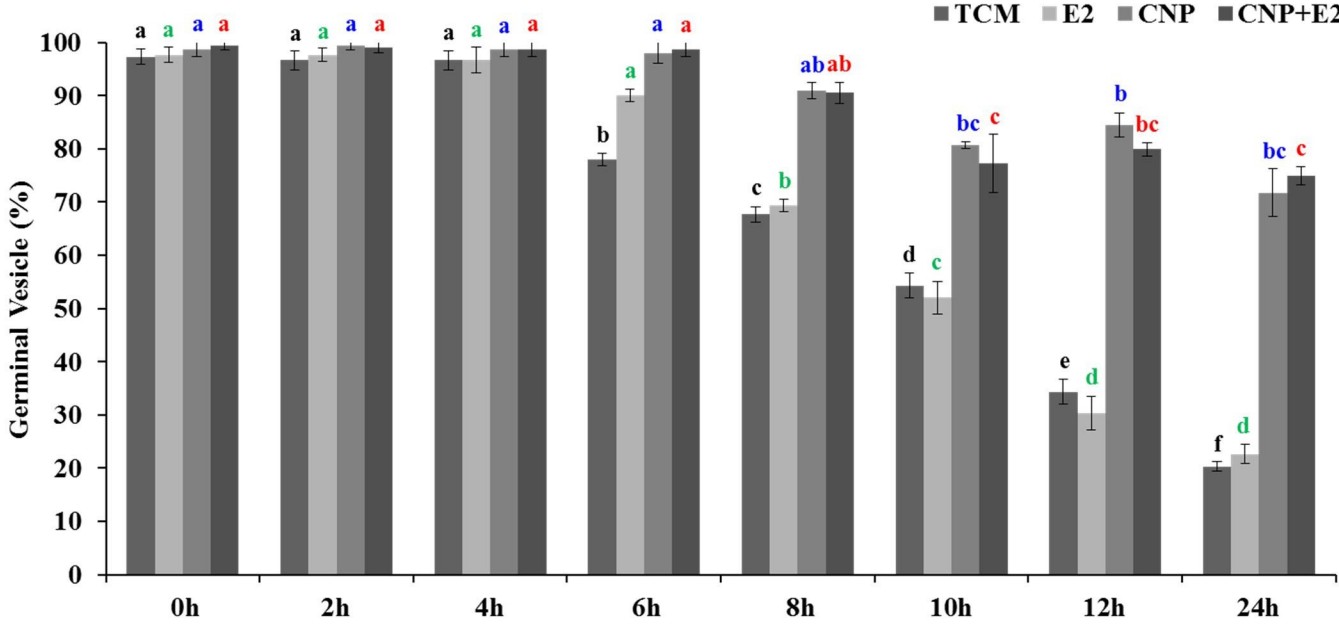

**B**
**Small size follicles**

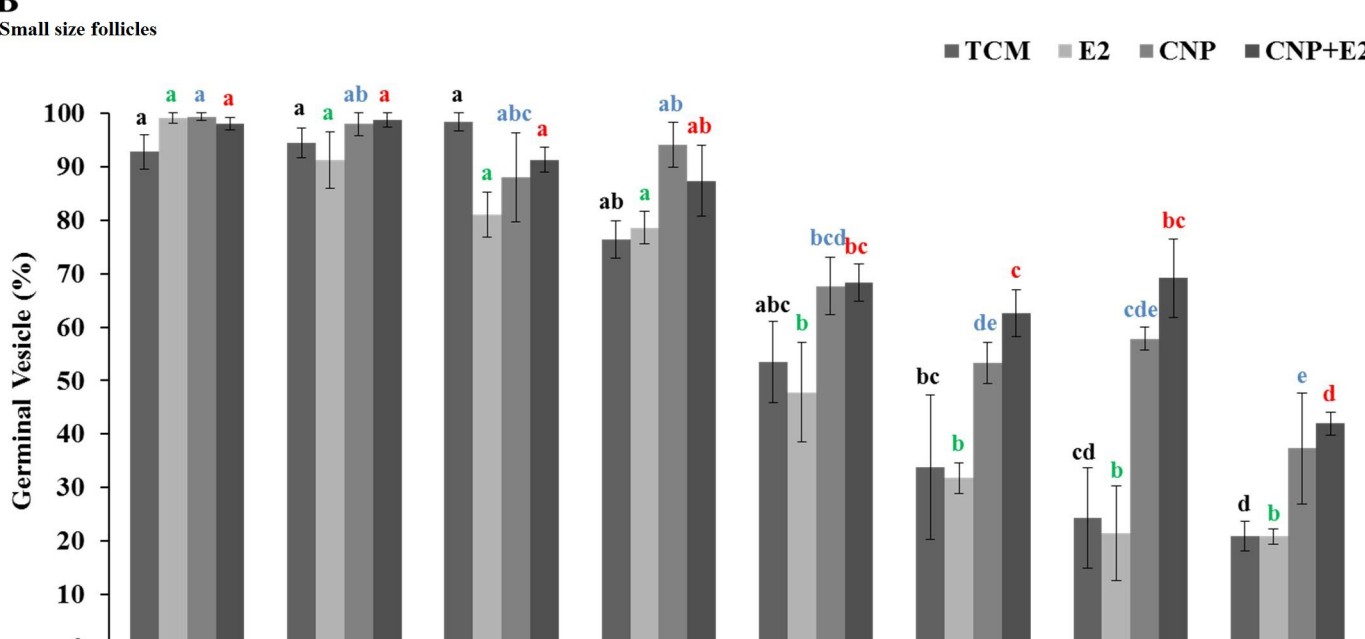

**Fig 5. Percentage of oocytes at the GV stage during 24 h IVM in four experimental groups (TCM, E2, CNP, CNP+E2). A**: medium size follicles (>4 to 6 mm)
**B**: small size follicles (2 to ≤4 mm). 3 replicates and the minimum number of oocytes in each replicate were 20. Different superscripts demonstrate significant differences (P<0.05) within columns with the same color. TCM: tissue culture medium, E2: estradiol, CNP: natriuretic peptide type C.

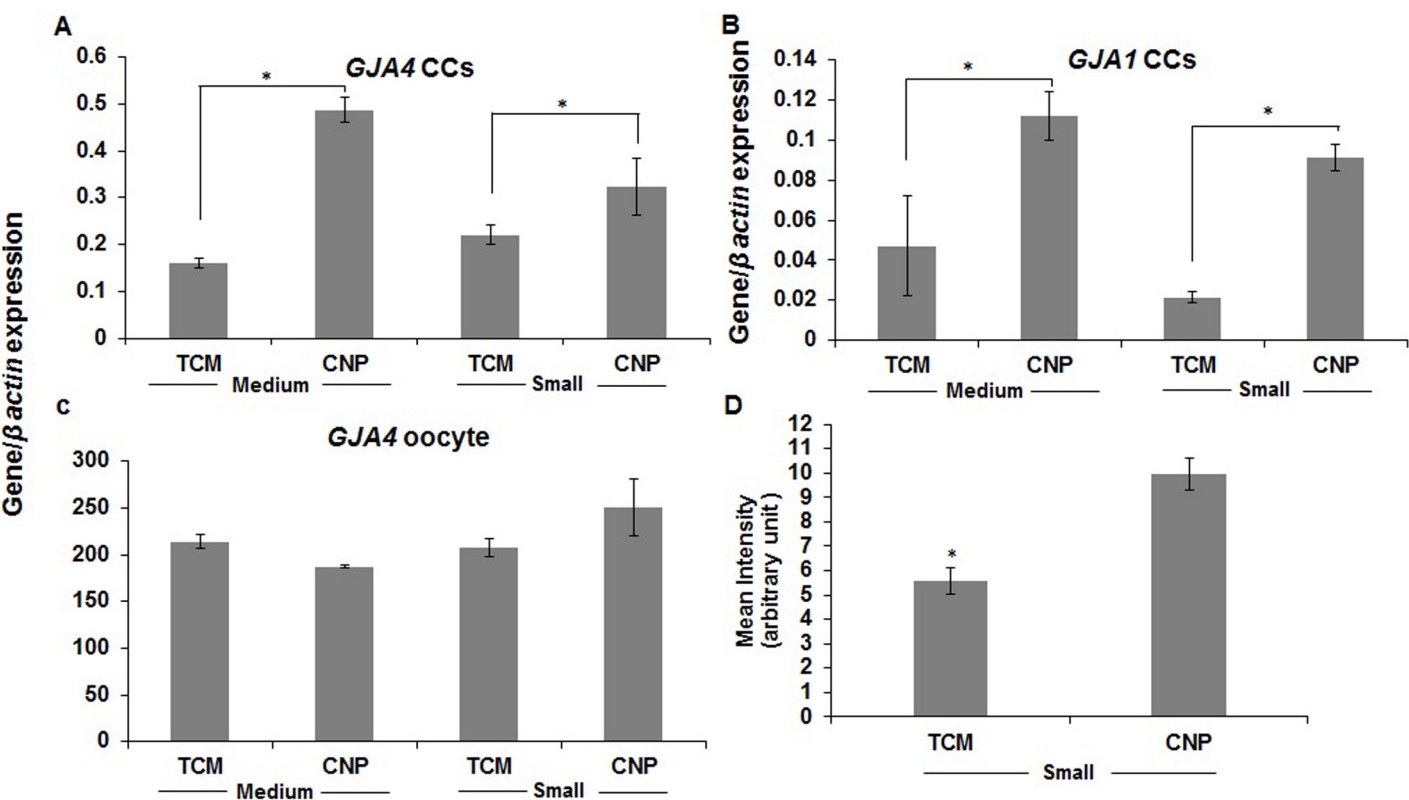

**Fig 6. A, B, C**: Gene expression of *GJA4* and *GJA1* between TCM and CNP groups in medium (>4 to 6 mm) and small (2 to ≤4 mm) size follicles. 3 replicates and the minimum number of oocytes and cumulus cells in each replicate were 30 and $2 \times 10^5$ cells, respectively. **D**: Calcein uptake through gap junctions between TCM and CNP groups in COCs derived from small size follicles (2 to ≤4 mm). 3 replicates and the minimum number of oocytes in each replicate were 10. The asterisks represent a significant difference. GJA: gap junction protein alpha, TCM: tissue culture medium, CNP: natriuretic peptide type C.

stage after 24 h IVM was 20.3 ± 0.8 in TCM (Fig 5A). In this group, in the presence of CNP (1000 nM), the percentage of oocytes arrested at the GV stage after 24 h IVM was 71.7 ± 4.4 [Fig 5A, see the experimental design (Fig 1, part II)].

Meiotic progression of oocytes derived from small size follicles occurred around 8 h after the onset of IVM in TCM. The percentage of oocytes derived from small size follicles at the GV stage after 24 h IVM was 20.9 ± 2 in TCM (Fig 5B). In this group, in the presence of CNP (100 nM), the percentage of oocytes arrested at the GV stage after 24 h IVM was 37.3 ± 10.3 [Fig 5B, see the experimental design (Fig 1, part II)]. Our results also showed E2 independently or in the presence of CNP did not affect meiotic arrest (Fig 5A and 5B).

Both gene expression of gap junction protein alpha 1 and 4 (*GJA1, GJA4)* in cumulus cells and calcein uptake through gap junctions in COCs derived from small and medium size follicles increased in the presence of CNP. But in the oocyte, *GJA4* gene expression in the presence of TCM or CNP in each group did not show any significant difference [Fig 6, see the experimental design (Fig 1, part II)].

## Developmental competence of COCs in the presence of AREG and PGE2

Optimal concentration of AREG in COCs derived from small size follicles (2 to ≤4 mm) was 300 nM because of the highest CEI (1.93), rate of M II (46.453 ± 0.73), cleavage (75.7± 3.9) and blastocyst rates (41.5± 1.5) (Table 2) (P<0.05).

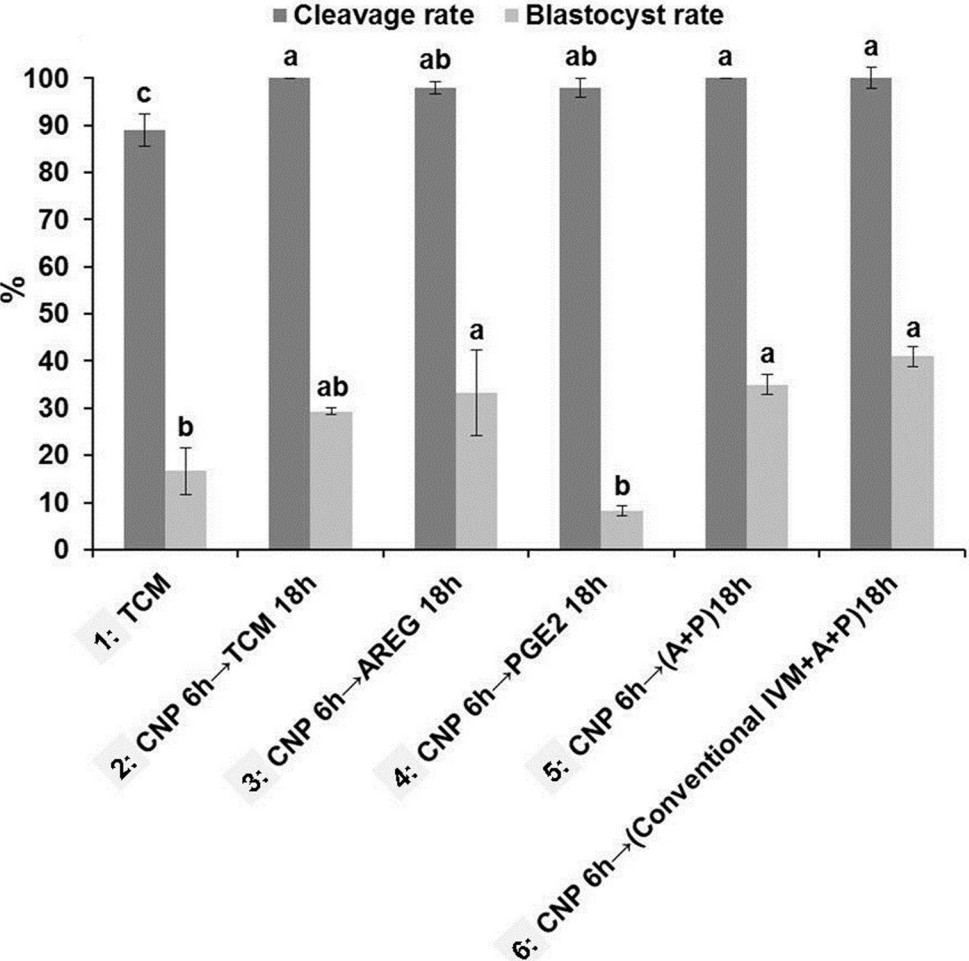

**Fig 7. Developmental competence of COCs harvested from small size follicles (2 to ≤4 mm) in sheep sequentially exposed to CNP for 6 h, then cultured in TCM+PGE2 and/or AREG for 18 h.** Different superscripts demonstrated significant differences (P<0.05) within columns with the same color. 5 replicates and the minimum number of oocytes in each replicate were 50. TCM: tissue culture medium, CNP: natriuretic peptide type C, AREG: amphiregulin, PGE2: prostaglandin E2.

Optimal concentration of PGE2 in COCs derived from small size follicles (2 to ≤4 mm) was 10 μM because of the highest rate of blastocyst (38.40 ± 0.51), but M II (%) (39.1 ± 1.6) did not show significant difference in comparison with 0.1 μM (36.4 ± 4.1) and 1 μM (31.0 ± 4.0) PGE2 (P>0.05). Also, the cleavage rate was not different between groups. The highest CEI was achieved in the presence of 1 μM PGE2 (P<0.05) (Table 3).

## Developmental competence of sequential COCs exposure to CNP, AREG and/or PGE2

To achieve this aim, 6 experimental groups were designed [Fig 7, see the experimental design (Fig 1, part III)]. Sequential exposure of COCs to group 3 [TCM+CNP (6 h), then cultured in TCM+AREG (18 h)], group 5 [TCM+CNP (6 h), then cultured in TCM+ AREG+PGE2 (18 h)], and group 6 [TCM+CNP (6 h), then cultured in conventional IVM supplements+AREG +PGE2 (18h)] showed higher blastocyst yield in comparison with TCM (P<0.05). But, group 4

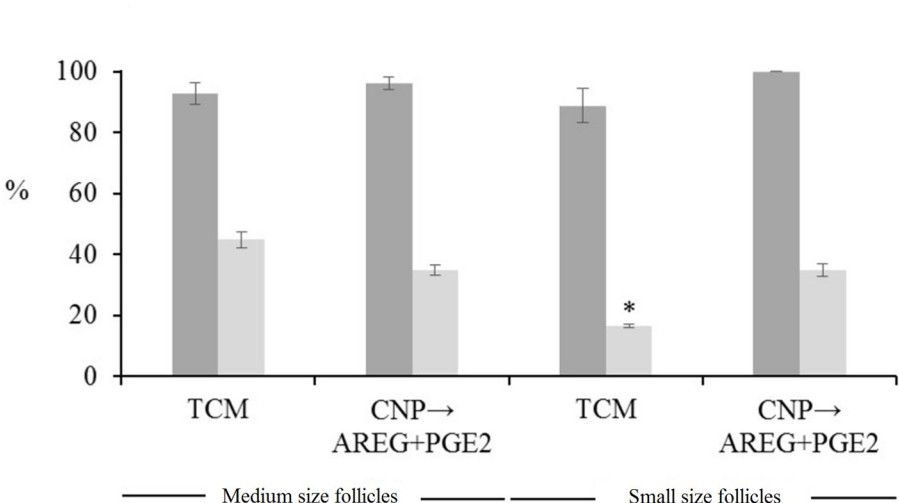

**Fig 8. *In vitro* development of COCs derived from medium (>4 to 6 mm) compared to small (2 to ≤4 mm) size follicles that cultured in TCM (control) or TCM+NP then cultured in TCM+AREG+PGE2 (treatment).** 5 replicates and the minimum number of oocytes in each replicate were 50. The asterisks represent the significant difference for blastocyst rate in small size follicles. TCM: tissue culture medium, CNP: natriuretic peptide type C, AREG: amphiregulin, PGE2: prostaglandin E2.

[TCM+CNP (6 h), then cultured in TCM+PGE2 (18 h)] showed lowest blastocyst yield in comparison with the three aforementioned (3, 5, 6) groups (P<0.05) (Fig 7).

Moreover, we checked for any differences between the [TCM+CNP (12 h), then cultured in TCM+AREG+PGE2 (12 h)] and [TCM+CNP (24 h), then cultured in TCM+AREG+PGE2 (24 h)] groups, but the development yields were not different for cleavage (97.56±5, 99±3.24) and blastocyst (26.22±1.5, 23±4.2) rates, respectively.

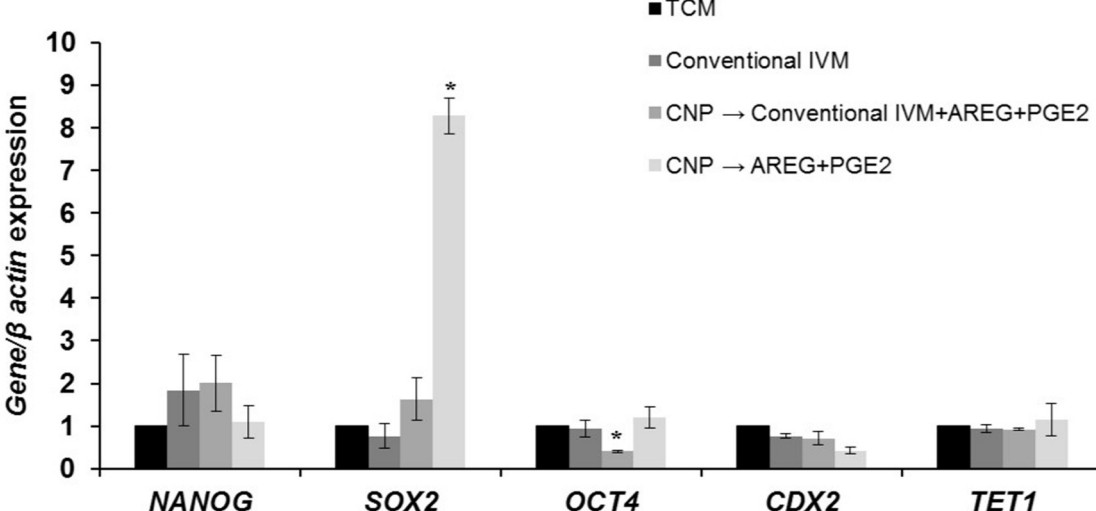

**Fig 9. Assessment of relative expression of pluripotency and epigenetic markers in blastocysts derived from TCM, conventional IVM (24 h), [TCM+CNP (6 h), then cultured in TCM+AREG+PGE2 (18 h)] and [TCM+CNP (6 h), then cultured in conventional IVM supplements + AREG + PGE2 (18 h)] groups.** 3 replicates and the minimum number of blastocysts in each replicate were 5. $2^{-(\text{delta delta CT})}$ was presented. The asterisks represent significant differences within columns with the same color. TCM: tissue culture medium, CNP: natriuretic peptide type C, AREG: amphiregulin, PGE2: prostaglandin E2.

**Table 2. Determination of optimal concentration of AREG in COCs derived from small size follicles.**

| Groups | CEI | M II (%) | Cleavage Rate (%) | Blastocyst Rate (%) |
|---|---|---|---|---|
| Control | 0.21[c] | 20.46 ± 2.11 [b] | 63.70 ± 5.03 | 16.35± 3.07 [b] |
| 50 nM AREG | 0.92 [b] | 42.043 ± 4.83 [ab] | 70.67 ± 2.22 | 21.63± 2.70 [b] |
| 100 nM AREG | 1.20[b] | 57.143 ± 9.18 [a] | 71.90 ± 3.83 | 29.06± 4.76 [ab] |
| 300 nM AREG | 1.93 [a] | 46.453 ± 0.73 [a] | 75.71± 3.96 | 41.52± 1.51 [a] |

CEI: cumulus expansion index, M II (%): Percentage of oocytes at the M II stage. Cleavage and Blastocyst rates were assessed by *in vitro* fertilization in COCs. Different superscripts demonstrate significant differences ($P < 0.05$) within each column. 5 replicates and the minimum number of oocytes in each replicate were 50, 20 and 20 for assessment of development, MII rate and CEI, respectively.

Also, CEI, cleavage and blastocyst rates of conventional IVM (24 h), [TCM+CNP (6 h), were compared, then cultured in TCM+AREG+PGE2 (18 h)] and [TCM+CNP (6 h), then cultured in conventional IVM supplements + AREG + PGE2 (18 h)]. It was seen that COCs in [TCM+CNP then cultured in conventional IVM supplements + AREG + PGE2] had higher CEI (3.56) and blastocyst rate (41±2.51) compared to conventional IVM (3.1 and 35.9± 5.0) and [TCM+CNP then cultured in TCM+AREG+PGE2] (3 and 34.9 ± 2.1) ($P > 0.05$).

[TCM+CNP (6 h), then cultured in TCM+AREG+PGE2 (18 h)] could only improve the development of oocyte derived from small follicles ($P < 0.05$) [Fig 8, see the experimental design (Fig 1, part III)]. In order to investigate whether [TCM+CNP (6 h), then cultured in TCM+AREG+PGE2 (18 h)] has the same effect on medium size follicles, we compared this treatment with TCM for medium size follicles. The results revealed no significant effect on cleavage and blastocyst rates in medium size follicles.

## Assessment of pluripotency and epigenetic markers in blastocysts derived from small size follicles

The only significant difference in gene expression was for *SOX2* which increased in the [TCM+CNP (6 h), then cultured in TCM+AREG+PGE2 (18 h)] group and for *OCT4 which* decreased in the [TCM+CNP (6 h), then cultured in conventional IVM supplements + AREG + PGE2 (18 h)] group ($P < 0.05$). *NANOG*, *CDX2*, and *TET1* did not show any significant differences among the groups ($P > 0.05$) [Fig 9, (see Fig 1, part III)].

## Discussion

One of the limiting factors in the application of IVM is a heterogeneous stage of follicles developing within or between ovaries. The heterogeneous populations of COCs derived from one ovary or different ovaries or from different abattoir have various developmental capacities or

**Table 3. Determination of optimal concentration of PGE2 in COCs derived from small size follicles.**

| Groups | CEI | M II (%) | Cleavage Rate (%) | Blastocyst Rate (%) |
|---|---|---|---|---|
| Control | 0.21[c] | 21.07 ± 1.02 [b] | 68.79 ± 2.27 | 20.16 ± 2.94 [b] |
| 0.1 µM PGE2 | 1.22 [b] | 31.05 ± 4.07 [ab] | 73.15 ± 2.87 | 32.53 ± 8.3 [b] |
| 1 µM PGE2 | 2.20[a] | 36.4 ± 4.1 [a] | 75.18 ± 2.58 | 30.67 ± 0.21 [b] |
| 10 µM PGE2 | 1.42 [b] | 39.18 ± 1.63 [a] | 76.96 ± 3.2 | 38.40 ± 0.51 [a] |

CEI: cumulus expansion index, M II (%): Percentage of oocytes at the M II stage. Cleavage and Blastocyst rates were assessed by *in vitro* fertilization in COCs. Different superscripts demonstrate significant differences ($P < 0.05$) within each column. 5 replicates and the minimum number of oocytes in each replicate were 50, 20 and 20 for assessment of development, MII rate, and CEI, respectively.

outcomes [10, 38]. Differences in morphology, adenosine triphosphate (ATP) content, metabolism, mitochondria distribution, protein, mRNA pattern, and methylation lead to different outcomes in *in vitro* production of the embryo (IVP) [10, 39]. Consistent with previous reports, in the present study, a reduction was observed in the developmental potential of ovine oocytes derived from small (2 to ≤ 4 mm) compared to medium (>4 to 6 mm) follicles, concluding that the follicular size affects developmental competency of COCs.

It is well defined that the isolation of COCs from the natural follicular environment results in spontaneous meiotic progression [40]. Granulosa cells under physiological conditions regulate the expression of transcripts like *NPPC/NPR2*, *PTGER2 /PTGS2*, *AREG/EGFR* involved in oocyte maturation, cumulus expansion, and ovulation through autocrine or paracrine mediators rather than circulating hormones [10, 28]. As we have shown, expression of these transcripts, especially *PTGER2 /PTGS2*, is lower in CCs isolated from small size follicles compared to medium ones. Also, our data showed that the relative *NPR2* transcript level decreased by passing of time during maturation in CCs derived from small follicles. Thus, the addition of these factors during IVM may improve IVP outcomes.

When CNP binds to guanylyl cyclase (GC)-coupled NPR2, cGMP production increases, and oocytes are maintained at the GV stage [41]. Therefore, it is not surprising to see that the CNP or Npr2 knockout rodent model results in an early resumption of meiosis, concluding the role of CNP in meiotic arrest. Similar functions for NPs were reported in mice, human, goat, cattle, pig, sheep, rat, and cat [42, 43]. In summary, the higher mRNA abundance of NP receptors in the dominant or medium follicles compared to the subordinate or smaller size follicles may be used as an indicator of follicle health and suggests that NPs signaling may regulate steroidogenesis and/or cell proliferation and differentiation [44]. The present study showed that the addition of 1000 nM and 100 nM CNP could arrest the majority of COCs derived from medium and small follicles for 24 h and 12 h, respectively. Recently, some studies have shown 6 h of culture with CNP (100 nM) can enhance oocyte development in cattle and goat [45, 46]. Furthermore, markers of gap junction communication (*GJA4* and *GJA1*) increased in the presence of CNP. Moreover, based on literature, chromatin configuration changes from non-surrounded nucleolus (NSN) to surrounded nucleolus (SN) during oocyte maturation, and this is regulated *in vitro* by NPs [47, 48]. Recently, a novel mechanism for the CNP-induced oocyte meiotic arrest has been introduced in bovine. Based on these results, bovine oocytes have NPR2 receptors and can mediate meiotic arrest [23]. Furthermore, inhibition of meiotic resumption with NPs instead of synthetic reagents like forskolin, PDE inhibitors and specific inhibitors of cyclin-dependent kinases (CDKs) and meiosis promoting factor (MPF) have been shown to have better outcomes as they maintain gap junction activity, and also support key gene expression, which are critical for oocyte development [49–51]. In addition, it has been stated that E2 can mediate NPR2 gene expression and may affect meiotic arrest [22]. Our results showed E2 independently or in the presence of CNP did not affect the meiotic arrest.

On the other hand, the final stage of oocyte maturation and ovulation is mediated by EGF-like peptides after surging with gonadotrophins. The mRNA levels of CNP in granulosa and Npr2 in cumulus cells reduce after LH/hCG treatment despite stimulation of EGF-like factors and activation of EGFR [44, 52]. But, according to our observation and other studies, the level of EGFR transcript decreases in the final stage of *in vitro* maturation in CCs derived from small size follicles. Therefore, it has been stated that the addition of AREG during maturation enhances bovine and porcine oocyte developmental competence [17, 42].

In ovulating follicles, the EGFR signaling cascade involves many other signaling networks in cumulus and granulosa cells, which participate in the development of oocyte competence. Follicle-stimulating hormone (FSH) mediates the induction of *AREG* mRNA via P38 mitogen-

activated protein kinases (p38MAPK). AREG also induces *PTGS2* expression via ERK1/2. PGs, also acting via PTGER2 in cumulus cells, provide a secondary, autocrine pathway to regulate expression of *AREG* in COCs. PGE2 acts on a group of G-protein-coupled receptors and can maintain high cAMP levels. PGE2 in our study resulted in mild cumulus expansion, similar to cow [53]. Previous findings indicate that the addition of PGE2 during IVM improves embryonic cell survival of blastocysts and post-hatching development. In this regard, *PTGS2⁻/⁻* mice present severe failure in the expansion of cumulus cells and extrusion of the first polar body. Our results also indicated that the addition of PGE2 improves cumulus expansion, percentage of MII oocytes, and developmental competency. Besides, we showed that all three factors (AREG, PGE2, and CNP) independently improved the development of COCs derived from small size follicles with TCM as a base medium [8, 14, 17, 43].

We also revealed that sequential exposure of ovine COCs to CNP then to AREG and/or PGE2 improved rate of blastocyst formation in [TCM+CNP, then cultured in TCM+AREG], [TCM+CNP, then cultured in TCM+AREG+PGE2], and [TCM+CNP, then cultured in conventional IVM supplements + AREG + PGE2] compared to TCM, with the best result observed in the latter group. Amongst the treatment groups, the latter also showed higher CEI compared to conventional IVM and [TCM + CNP, then cultured in TCM+AREG+PGE2].

Expansion of the extra-cellular matrix during oocyte maturation is essential for acquiring molecular machinery competence *in vivo* but *in vitro* overexpansion by promoting the hexosamine biosynthesis pathway (HBP) negatively influences *in vitro* oocyte competence [47, 54–56]. Therefore, despite higher CEI in this group, super developmental capacity was not observed.

Expression of embryo quality markers at RNA level revealed that their patterns were not different between conventional IVM, [TCM+CNP, then cultured in TCM+AREG+PGE2], and [TCM+CNP, then cultured in conventional IVM supplements + AREG+PGE2] for *NANOG*, *CDX2*, and *TET1*. But *SOX2* significantly increased in [TCM+CNP (6 h), then cultured in AREG+PGE2], and *OCT4* decreased in [TCM+CNP (6 h), then cultured in conventional IVM supplements + AREG + PGE2]. In mammals, despite specific species differences, the transcription of triad factors (NANOG, *SOX2*, and OCT4) is governed through feedback loops in a steady state. Therefore, knock-down of one factor results in the up-regulation of one or two of the other triad factors [57]. In the studies assessing temporal expression of the triad factors in ruminants, it was shown that the temporal expression of triads is stage-specific dependent, and the expression of *OCT4* is induced during oocyte maturation and declines following development to blastocyst stage, while *SOX2* and *NANOG* are transcribed during maternal/zygote transition and among triads, *SOX2* presents the highest expression relative to references gene [58]. Therefore, the high expression of *SOX2* in the treated group is consistent with the literature, and higher expression of *SOX2* suggests improved zygote genomics activation in the treated group. Further emphasized, improved *in vitro* maturation in this group reflects itself in an improved effect on *SOX2* expression. But, these propositions need further experiments and verifications. One of the shortcomings of this study is that we did not perform concomitant differential staining to observe whether this differential expression may have any effect on total cell numbers and cell allocations.

## Conclusion

Despite interspecies differences among sheep with human and other farm animals, the sequential maturation of sheep oocyte by stimulating NP/NPR2, AREG/EGFR, and PGE2/PTGSR2 system can improve the quality of COCs that have reduced expression of these functional pathways, meaning that COCs derived from small size follicles are less competent due to lower

expression of above genes. Therefore, based on the results of this study and others, the supplementation of IVM medium with products of COCs [59] and GCs secreted factors improves developmental capacity and is an alternative to synthetic chemical treatment.

## Acknowledgments

We thank Professor Jeremy G Thompson, The University of Adelaide, for his helpful assistance suggesting the study and assistance with editing.

## Author Contributions

**Investigation:** Mehdi Hajian, Farnoosh Jafarpour.

**Methodology:** Shiva Rouhollahi Varnosfaderani, Faezeh Ghazvini Zadegan.

**Supervision:** Mohammad Hossein Nasr-Esfahani.

**Writing – original draft:** Shiva Rouhollahi Varnosfaderani.

**Writing – review & editing:** Shiva Rouhollahi Varnosfaderani.

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
