## [Decision Letter · Decision Letter 0]

15 Oct 2019

PONE-D-19-25159

Granulosa secreted factors improve developmental competence of cumulus-oocyte complexes from small antral follicles

PLOS ONE

Dear Dr nasr esfahani,

Thank you for submitting your manuscript to PLOS ONE. After careful consideration, we feel that it has merit but does not fully meet PLOS ONE’s publication criteria as it currently stands. Therefore, we invite you to submit a revised version of the manuscript that addresses the points raised during the review process.

In addition to the concerns regarding the content, pay particular attention to the English. Write in full sentencences instead using symbols (for example the used arrows, ll 34-38, but also ll358-380). Please mention the species in the title, and avoid abbreviations  in the Abstract.

We would appreciate receiving your revised manuscript by Nov 29 2019 11:59PM. To enhance the reproducibility of your results, we recommend that if applicable you deposit your laboratory protocols in protocols.io, where a protocol can be assigned its own identifier (DOI) such that it can be cited independently in the future. For instructions see: http://journals.plos.org/plosone/s/submission-guidelines#loc-laboratory-protocols

We look forward to receiving your revised manuscript.

Kind regards,

Wilfried A. Kues, Ph.D.

Academic Editor

PLOS ONE

Journal Requirements:

2. In your Methods section, please provide additional details regarding the sperm used in your study and ensure you have described the source. For more information regarding PLOS' policy on materials sharing and reporting, see https://journals.plos.org/plosone/s/materials-and-software-sharing#loc-sharing-materials.

Additional Editor Comments (if provided):

Reviewers' comments:

Reviewer's Responses to Questions

**Comments to the Author**

1. Is the manuscript technically sound, and do the data support the conclusions?

Reviewer #1: Partly

2. Has the statistical analysis been performed appropriately and rigorously? 

Reviewer #1: No

3. Have the authors made all data underlying the findings in their manuscript fully available?

Reviewer #1: Yes

4. Is the manuscript presented in an intelligible fashion and written in standard English?

Reviewer #1: No

5. Review Comments to the Author

Reviewer #1: In the manuscript (granulosa secreted factors improve developmental competence of cumulus-oocyte complexes from small antral follicles - PONE-D-19-25159) by Shiva R. Varnosfaderani et al., the authors assess IVM in sheep oocytes originating from follicles of different sizes, and the effects of CNP, AREG and PGE2 on the development of those oocytes. Findings reported in this manuscript are of interest, but several major concerns have been identified.

Comments:

1. Experimental groups should be renamed and better defined to make figures cleaner and easier to understand. For example: ‘small’ and ‘large’ instead of ‘2 to ≤ 4mm’ and ‘≥ 4 to 6 mm’.

2. There are a considerable amount of grammatical and syntax errors throughout the manuscript. Pay special attention to the tense and plurality of words and phrases.

3. It is stated in the materials and methods section that ovaries were collected in saline and stored for an additional 12 hours at 15 °C. However, no explanation or purpose for this resting period and exposure to low temperatures was given.

4. Figure 1 showing the experimental design is helpful since it allows to the reader to understand what was done quickly. However, the figures referenced in it, do not match the figure order in the manuscript, and groups should be renamed to make it easier to understand.

5. In Figure 2, you show the cleavage and blastocyst rate among follicle sizes. Cleavage rates are shown to be over 90%, and yet later in a subsequent experiment, it is shown that maturation rates are below 80%. How do you account for these incongruences? Were non-mature oocytes removed in figure 2?

6. The groupings in figures should be consistent, with small follicles always on the left and large follicles always on the right, since, as it stands at the moment, small are sometimes on the left and sometimes on the right, and vice-versa.

7. The scales in the figures are also inconsistent, for example, in figure 5, the scale goes to 120%, this is redundant as it 100% would be the maximum.

8. In figure 3, what is the purpose of showing zero hour exposure? This should simply be the control.

9. In figure 4, the groupings and methodology for those groupings are unclear and poorly defined. How can you compare different CNP concentrations among follicle sizes, if they were matured for different time lengths? Furthermore, explicit explanation for this must be defined about this within the text.

10. In figure 5, it is impossible to compare small and large follicles since according to figure 4, they were matured for different lengths, with different CNP concentrations.

11. It is unclear what the differences between ‘CNP 6h � (A + P)18h’ and ‘CNP 6h � (Conventional IVM + A + P)18h’ are.

12. In Figure 9, why are there no error bars in the control TCM group? The huge change in the SOX2 expression in the ‘CNP � AREG + PGE2’ group must be explained and considered with more depth in the discussion section.

13. The materials and methods section is also unclear about specific experiments. What was the n in each experiment? How was blastocyst rate calculated (over cleavage or over oocyte?), were non-mature oocytes removed?

14. The discussion section is long and a little unfocused. For example, the first paragraph talks about IVM and how it relates to ART in human medicine, which is redundant in a manuscript using the ovine model.

6. PLOS authors have the option to publish the peer review history of their article (what does this mean?). If published, this will include your full peer review and any attached files.

Reviewer #1: No

---

## [Author Response · Author response to Decision Letter 0]

2 Dec 2019

Dear Dr. Wilfried A. Kues,

Hereby, we are submitting the revised version of our manuscript entitled: “Granulosa secreted factors improve developmental competence of cumulus-oocyte complexes from small antral follicles” (PONE-D-19-25159). We would like to thank you and respected reviewer for their valuable comments. Below are our responses to the editor's and reviewers’ comments. The original comments of the Editor's and reviewers are in black and our responses are in blue. For better check out, the implemented changes were defined as underlined in the revised manuscript. We hope the changes are satisfactory.

Best Regards

M. H. Nasr-Esfahani, 

Email: mh.nasr-esfahani@royaninstitute.org

Editor comments:

In addition to the concerns regarding the content, pay particular attention to the English. Write in full sentencences instead using symbols (for example the used arrows, ll 34-38, but also ll358-380). Please mention the species in the title, and avoid abbreviations in the Abstract.

Reply: 

Thanks for your comments; the manuscript was edited by a native English speaker. We hope the substantial changes are satisfactory. 

The name of sheep has been added to the title of the article. 

Abbreviations were removed from the abstract. 

The symbols were also removed in the text. 

All above changes were underlined in the text.

Journal Requirements:

Reply: The PLOS ONE's style is implemented we hope the changes are satisfactory.

2. In your Methods section, please provide additional details regarding the sperm used in your study and ensure you have described the source. For more information regarding PLOS' policy on materials sharing and reporting, see https://journals.plos.org/plosone/s/materials-and-software-sharing#loc-sharing-materials.

Reply: Details were added to the Methods section (L: 112, 138, 163, 185). 

Reply: The information for this section was added (L: 290, 293). 

Reviewer comments:

Reviewer #1: In the manuscript (granulosa secreted factors improve developmental competence of cumulus-oocyte complexes from small antral follicles - PONE-D-19-25159) by Shiva R. Varnosfaderani et al., the authors assess IVM in sheep oocytes originating from follicles of different sizes, and the effects of CNP, AREG and PGE2 on the development of those oocytes. Findings reported in this manuscript are of interest, but several major concerns have been identified.

Thanks, we have addressed the issues raised (statistical analysis section, groups name and English) and the paper have modified accordingly. We hope the changes are satisfactory.

1. Experimental groups should be renamed and better defined to make figures cleaner and easier to understand. For example: ‘small’ and ‘large’ instead of ‘2 to ≤ 4mm’ and ‘≥ 4 to 6 mm’.

Reply: Experimental groups renamed in the text. The symbols (�) were also removed in the text. Also, ‘2 to ≤ 4mm’ and ‘≥ 4 to 6 mm’ defined to ‘small’ and ‘medium’. All changes were underlined in the text.

2. There are a considerable amount of grammatical and syntax errors throughout the manuscript. Pay special attention to the tense and plurality of words and phrases.

Reply: Grammatical, syntax errors, tense, plurality of words and phrases throughout the manuscript was edited to best of our ability and we hope the changes is satisfactory. 

 3. It is stated in the materials and methods section that ovaries were collected in saline and stored for an additional 12 hours at 15 °C. However, no explanation or purpose for this resting period and exposure to low temperatures was given.

Reply: L:112-114 was rewritten and please see the below reason, we provided a reference for this.

In our country, local slaughterhouses (Fasaran slaughterhouse in Isfahan, Iran) get started at 2:00 PM and store meat in refrigerator for 12 hours to complete rigor mortis. Thus, ovaries are transported and arrived into the lab at 6:00 PM. Since the lab working hours starts in the morning, they are stored at 15 °C for 12 hours. It is worth noting that this temperature and the duration are set up in our lab as the best conditions for maintenance of ovaries based on pervious literature (ref:32,33).

4. Figure 1 showing the experimental design is helpful since it allows to the reader to understand what was done quickly. However, the figures referenced in it, do not match the figure order in the manuscript, and groups should be renamed to make it easier to understand.

Reply: Thank you for your precision. We have checked and corrected the order. Please see Figure 1.

5. In Figure 2, you show the cleavage and blastocyst rate among follicle sizes. Cleavage rates are shown to be over 90%, and yet later in a subsequent experiment, it is shown that maturation rates are below 80%. How do you account for these incongruences? Were non-mature oocytes removed in figure 2?

Reply: Thanks for your precision. Please note that for IVM (Figure 2) the medium contain all the ingredients or supplements to show that small follicles are less component than large follicles. While in the remaining experiments, the IVM medium is simple TCM and one or two or more component of the supplements. Therefore, lower maturation rate is not unexpected. To make the point easier for readers the text (group’s name) was modified. 

 6. The groupings in figures should be consistent, with small follicles always on the left and large follicles always on the right, since, as it stands at the moment, small are sometimes on the left and sometimes on the right, and vice-versa.

Reply: Thank you for your precision. This note was corrected in Figure 6 and the text.

 7. The scales in the figures are also inconsistent, for example, in figure 5, the scale goes to 120%, this is redundant as it 100% would be the maximum.

Reply: Thanks, this note was corrected.

 8. In figure 3, what is the purpose of showing zero hour exposure? This should simply be the control.

Reply: To show the trend of mRNA changes over the IVM period, 2^-(delta CT) was presented. Please see the changes in the text (L: 163).

 9. In figure 4, the groupings and methodology for those groupings are unclear and poorly defined. How can you compare different CNP concentrations among follicle sizes, if they were matured for different time lengths? Furthermore, explicit explanation for this must be defined about this within the text.

Reply: Thanks, yes, we did not compare the small and large follicles. For better readout we separated the groups in Fig 4.

 10. In figure 5, it is impossible to compare small and large follicles since according to figure 4, they were matured for different lengths, with different CNP concentrations.

Reply: Yes, you are correct and we did not compare the small and large follicles. But to make the point clear we have modified the text (L: 238-248).

 11. It is unclear what the differences between ‘CNP 6h � (A + P)18h’ and ‘CNP 6h � (Conventional IVM + A + P)18h’ are.

Reply: Conventional IVM contains TCM + LH (10µg/ml) + FSH (10µg/ml) + IGF (100ng/ml) + EGF (10ng/ml) + Cys (0.1mM) + FBS (15%) + E2 (1µg/ml). For better readout of the manuscript this information was added where appropriate (Figure1, L: 203). 

 12. In Figure 9, why are there no error bars in the control TCM group? 

Reply: Thanks, as we used the formula 2^-(delta delta CT) and the data is normalized with control (TCM) therefore, our control group (TCM) would always one and therefore, the error bar would be zero, that cannot be shown.

Compared the expression of the target genet compared to the huge change in the SOX2 expression in the ‘CNP � AREG + PGE2’ group must be explained and considered with more depth in the discussion section.

Reply: Thanks for your constructive comment. We search the literature and we believe we reach a better explanation. Therefore, that paragraph was removed and a new paragraph was added.

 13. The materials and methods section is also unclear about specific experiments. What was the n in each experiment? How was blastocyst rate calculated (over cleavage or over oocyte?), were non-mature oocytes removed?

Reply: Numbers were added to figure legends. Blastocysts rate was over cleavage and added in the text (L: 152). Details were added to the Methods section (L: 112, 138, 163, 185).

 14. The discussion section is long and a little unfocused. For example, the first paragraph talks about IVM and how it relates to ART in human medicine, which is redundant in a manuscript using the ovine model.

Reply: Discussion was shortened and unnecessary information (first paragraph) was deleted. All changes were underlined in the text.

---

## [Decision Letter · Decision Letter 1]

18 Dec 2019

PONE-D-19-25159R1

Granulosa secreted factors improve developmental competence of cumulus-oocyte complexes from small antral follicles in sheep

PLOS ONE

Dear Dr nasr esfahani,

Thank you for submitting your manuscript to PLOS ONE. After careful consideration, we feel that it has merit but does not fully meet PLOS ONE’s publication criteria as it currently stands. Therefore, we invite you to submit a revised version of the manuscript that addresses the points raised during the review process.

We would appreciate receiving your revised manuscript by Feb 01 2020 11:59PM. To enhance the reproducibility of your results, we recommend that if applicable you deposit your laboratory protocols in protocols.io, where a protocol can be assigned its own identifier (DOI) such that it can be cited independently in the future. For instructions see: http://journals.plos.org/plosone/s/submission-guidelines#loc-laboratory-protocols

We look forward to receiving your revised manuscript.

Kind regards,

Wilfried A. Kues, Ph.D.

Academic Editor

PLOS ONE

Reviewers' comments:

Reviewer's Responses to Questions

**Comments to the Author**

1. If the authors have adequately addressed your comments raised in a previous round of review and you feel that this manuscript is now acceptable for publication, you may indicate that here to bypass the “Comments to the Author” section, enter your conflict of interest statement in the “Confidential to Editor” section, and submit your "Accept" recommendation.

Reviewer #1: (No Response)

2. Is the manuscript technically sound, and do the data support the conclusions?

Reviewer #1: Yes

3. Has the statistical analysis been performed appropriately and rigorously? 

Reviewer #1: Yes

4. Have the authors made all data underlying the findings in their manuscript fully available?

Reviewer #1: Yes

5. Is the manuscript presented in an intelligible fashion and written in standard English?

Reviewer #1: Yes

6. Review Comments to the Author

Reviewer #1: The revised manuscript has been significantly improved and all the major concerns were addressed.

Compared to the first version, major improvements have been made to the grammar making the manuscript much easier to read and understand. However, some minor grammar mistakes still persist.

7. PLOS authors have the option to publish the peer review history of their article (what does this mean?). If published, this will include your full peer review and any attached files.

Reviewer #1: No

---

## [Author Response · Author response to Decision Letter 1]

27 Jan 2020

Dear Dr. Wilfried A. Kues,

Hereby, we are submitting the revised version of our manuscript entitled: “Granulosa secreted factors improve the developmental competence of cumulus oocyte complexes from small antral follicles” (PONE-D-19-25159R2). 

We would like to thank you and respected reviewer for valuable comments. The original comment is in black and our response is in blue. For better check out, the implemented changes were defined as underlined in the revised manuscript. We hope it has now acquired the high status to meet the requirements for publication in your esteemed journal.

Best Regards

M. H. Nasr-Esfahani, 

Email: mh.nasr-esfahani@royaninstitute.org

Reviewer comment:

Reviewer #1: The revised manuscript has been significantly improved and all the major concerns were addressed. Compared to the first version, major improvements have been made to the grammar making the manuscript much easier to read and understand. However, some minor grammar mistakes still persist.

Reply: The authors appreciate the reviewer’s valuable comment. All grammar mistakes throughout the manuscript were carefully edited, and we hope the changes are satisfactory. All changes were underlined in the text.

---

## [Editor Report · Decision Letter 2]

29 Jan 2020

Granulosa secreted factors improve the developmental competence of cumulus oocyte complexes from small antral follicles in sheep

PONE-D-19-25159R2

Dear Dr. nasr esfahani,

We are pleased to inform you that your manuscript has been judged scientifically suitable for publication and will be formally accepted for publication once it complies with all outstanding technical requirements.

With kind regards,

Wilfried A. Kues, Ph.D.

Academic Editor

PLOS ONE
---

## [Editor Report · Acceptance letter]

3 Feb 2020

PONE-D-19-25159R2 

Granulosa secreted factors improve the developmental competence of cumulus oocyte complexes from small antral follicles in sheep 

Dear Dr. nasr esfahani:

I am pleased to inform you that your manuscript has been deemed suitable for publication in PLOS ONE. Congratulations! Your manuscript is now with our production department. 

With kind regards,

on behalf of

Dr. Wilfried A. Kues 

Academic Editor

PLOS ONE